# Research on Hyperparameter Optimization of Concrete Slump Prediction Model Based on Response Surface Method

**DOI:** 10.3390/ma15134721

**Published:** 2022-07-05

**Authors:** Yuan Chen, Jiaye Wu, Yingqian Zhang, Lei Fu, Yunrong Luo, Yong Liu, Lindan Li

**Affiliations:** 1School of Civil Engineering, Sichuan University of Science & Engineering, Zigong 643000, China; 17383502283@163.com (Y.C.); 320085607104@stu.suse.edu (L.L.); 2School of Mechanical and Electrical Engineering, Southwest Petroleum University, Chengdu 610500, China; wujy@scentralit.com; 3School of Mechanical Engineering, Sichuan University of Science & Engineering, Zigong 643000, China; 4Key Laboratory of Process Equipment and Control Engineering of Sichuan Province Colleges and Universities, Zigong 643000, China; yunrong_luo@suse.edu.cn; 5College of Chemical Engineering and Technology, Sichuan University of Science & Engineering, Zigong 643000, China; lldoyxy6999@163.com

**Keywords:** BP neural network, response surface method, parameter optimization, slump, determination coefficient, root mean square error

## Abstract

In this paper, eight variables of cement, blast furnace slag, fly ash, water, superplasticizer, coarse aggregate, fine aggregate and flow are used as network input and slump is used as network output to construct a back-propagation (BP) neural network. On this basis, the learning rate, momentum factor, number of hidden nodes and number of iterations are used as hyperparameters to construct 2-layer and 3-layer neural networks respectively. Finally, the response surface method (RSM) is used to optimize the parameters of the network model obtained previously. The results show that the network model with parameters obtained by the response surface method (RSM) has a better coefficient of determination for the test set than the model before optimization, and the optimized model has higher prediction accuracy. At the same time, the model is used to evaluate the influencing factors of each variable on slump. The results show that flow, water, coarse aggregate and fine aggregate are the four main influencing factors, and the maximum influencing factor of flow is 0.875. This also provides a new idea for quickly and effectively adjusting the parameters of the neural network model to improve the prediction accuracy of concrete slump.

## 1. Introduction

With the development of concrete technology and performance, high-performance concrete and self-compacting concrete have been widely used. These concrete constructions no longer simply consider the strength of the concrete, but also consider the durability and workability of the concrete. Concrete starts to be mixed in the mixing station until it is transported to the site for pouring, but it takes time for transportation and parking on the way, which makes the workability of the concrete worse. The slump loss of concrete directly affects the workability of concrete, causes difficulties for construction, may bring about construction accidents and affects the quality of hardened concrete. Therefore, analyzing the causes of excessive concrete slump has particular significance for preventing the loss of concrete slump, thereby improving the workability of concrete.

Regarding global construction material usage [1], concrete is one of the most in-demand and most adaptable materials. Slump is an important indicator to measure the uniformity of concrete quality. It is reflected in the fact that the slump slows down the hardening speed of concrete and causes the strength after hardening to be low, which greatly affects the quality of the project. The test procedure is shown in Figure 1 [2]. Precise control of slump [3,4] is a prerequisite for ensuring the excellent performance of concrete. In practical engineering, the slump measurement test often requires a lot of time, manpower and material resources, and it is difficult to obtain the test results quickly and accurately. Therefore, the study of a concrete slump prediction model [5,6] is of extraordinary significance for the theory and application of construction engineering.

With the advancement of science and technology and the development of the construction industry, civil engineering has put forward higher and higher requirements for the performance of concrete materials, and the traditional mix design method [7] largely relies on the experience of designers. Recently, civil and architectural researchers have gradually introduced artificial neural network [8], genetic algorithm [9] and other artificial intelligence sciences into the optimization design of concrete mix ratio and have achieved a series of scientific research results. X. Hu [10] et al. proposed an ensemble model for concrete strength prediction, and gave measures to improve its prediction accuracy, such as optimizing features, ensemble algorithms, hyperparameter optimization, expanding sample data sets, richer data sources and data preprocessing, etc. M. Shariati [11] et al. used a hybrid artificial neural network genetic algorithm (ANN-GA) as a new method to predict the strength of concrete slag and fly ash. The results show that the ANN-GA model can not only be developed and adapted to the compressive strength prediction of concrete, but it can also produce better results compared to the artificial neural network back propagation (ANN-BP) model. I-Yeh [12,13] described methods for predicting slump and compressive strength of high-performance concrete. Venkata [14] et al. proposed a feasibility assessment of strength properties of self-compacting concrete based on artificial neural networks. Duan [15] et al. used artificial neural network to predict the compressive strength of recycled aggregate concrete and achieved good results. Ji [16] et al. proposed an in-depth algorithmic study of concrete mix ratios using neural network. Demir [17] predicted the elastic modulus of ordinary and high-strength concrete by using artificial neural networks. Vinay [18] predicted ready-mixed concrete slump based on genetic algorithm. Li Dihong [19] et al. predicted the comprehensive properties of concrete with back propagation neural network. I-Cheng Yeh [20] and Ashu Jain [21] respectively proposed an optimal design method of concrete mix ratio based on artificial neural networks. Wang Jizong [22], Chul-Hyun Lim [23], Liu Cuilan [24], and I-Cheng Yeh [25] applied the genetic algorithm to the optimal design of concrete mix ratios.

To sum up, although the neural network has been widely used in civil engineering, it is relatively rare to apply it to concrete slump prediction and model parameter optimization. Therefore, this paper plans to use the BP neural network model to predict the slump of concrete and uses eight variables, such as cement, blast furnace slag, fly ash, water, superplasticizer, coarse aggregate, fine aggregate and flow as the network input, and slump as the network output. The learning rate, momentum factor, number of hidden nodes and the number of iterations are used as hyperparameters to build 2-layer and 3-layer neural networks. Finally, the parameters of the prediction model are studied through single factor and RSM in order to obtain a wider range of optimal parameter solutions. The research process of this paper is shown in Figure 2.

## 2. Establishment of a Concrete Slump Model Based on BP Neural Network

### 2.1. Introduction to BP Neural Network

Artificial neural networks, also known as neural networks (NNs), are composed of widely interconnected processing units. They can not only accurately classify [26,27] a large number of unrelated data but also have a good predictive function. The function between the input and output of each neuron is called the activation function, also known as the S-shaped growth curve. The structure of the neural network essentially imitates the structure and function of the human brain nervous system to establish a neural network model. It is a highly complex information processing system, and its network architecture is based on a specific mathematical model, as shown in Figure 2. The shown neural network architecture (take three layers as an example), including input layer, hidden layer and output layer [28,29,30]. Most neural networks are based on back propagation, and the back propagation training (back propagation: BP) algorithm adjusts the weights of neurons through the gradient descent method [31], the purpose is to minimize the error between the actual output and the expected output of the multilayer feedforward [32] neural network. In other words, until the mean square error of all training data is minimized to the specified error range.

The BP neural network learning algorithm can be described as the most successful neural network learning algorithm. When using neural networks in display tasks, most of them use BP algorithm for training. The network model in this paper contains the basic structure of the BP neural network, namely the input layer, hidden layer and output layer. The signal is activated through the input layer and then the features are extracted through the hidden layer. The neural units of different hidden layers may have different weights and self-biases corresponding to different input layers. The excitation of the input layer is transmitted to the hidden layer, and finally the output layer generates results according to different hidden layers, layer weights and self-bias. The algorithm flow chart of the BP neural network is shown in Figure 3. The activation function adopts the Sigmoid function, and the operation principle of the Sigmoid function is as follows:

The first is the linear feature normalization function:(1)$$X_{norm}=X-X_{min\;}/X_{max\;}-X_{min\;}$$

Linear normalization: Perform linear transformation on the original data to map the data between [0, 1] to achieve equal scaling of the data. The specific announcement is shown in formula (1). where *X* is the original data, *X**_min_* is the minimum value of the data, and *X**_max_* is the maximum value of the data.

The overall density of the probability is 1, and when the *X* vector is used as a constant exponential vector, the result is the inverse proportion of the whole function on the *Y* axis. So Sigmoid is defined as [33]:(2)$$\mathrm{Sigmoid}=\frac{1}{1+e^{-X}}$$

In this way, the probability of the vector can be normalized, and its value is between plus and minus 1.

The neurons in each layer of the network adopt the gradient descent momentum learning method, which is expressed as follows [34]:(3)$$d\omega =mc\times d\omega_{prev}+\left(1-mc\right)\times lr\times g\omega$$
where:

*dω* = the amount of change in the weight (threshold) of a single neuron;

*dω_prev_* = the amount of change in neuron weight (threshold) value in the previous iteration;

*mc* = momentum factor;

*lr* = learning rate;

*gω* = weight (threshold) gradient.

### 2.2. Samples and Network Input and Output

This test is a dataset composed of 103 sets of experimental data, and the data come from the network public test data of Prof. I-Cheng Yeh [35], including eight variables, such as cement, blast furnace slag, fly ash, water, superplasticizer, coarse aggregate, fine aggregate, flow (fluidity refers to the ability of the cement mortar mixture to generate fluidity and fill the mold uniformly and densely under the action of its own weight or mechanical vibration. Slump is one of the key indicators reflecting the flow properties of concrete. The two influence each other, and the flow of cement mortar affects the size of concrete slump. The introduction of the flow of the cement mortar makes the results of the model more convincing.). The concrete mix proportion is shown in Table 1 and Figure 4. Among them, cement, blast furnace slag, fly ash, water, superplasticizer, coarse aggregate, fine aggregate and flow are the inputs of neural network variables, and the slump is the output of variables. 

### 2.3. Network Data Preprocessing

Data preprocessing can help improve the quality of data, which in turn helps improve the effectiveness and accuracy [36] of the data mining process. The specific work includes the following:(1)Shuffle the data: Shuffling the data before model training can effectively reduce the variance [37,38] and ensure that the model remains general while reducing overfitting.(2)Outlier processing: It is inevitable that we will have a few data points that are significantly different from other observations during training. A data point is considered to be an outlier [39] if it lies 1.5 times the interquartile range below the first quartile or above the third quartile. The existence of outliers will have a serious impact on the prediction results of the model. Therefore, our processing of outliers [40,41] is generally to delete them. Figure 5 is a box diagram of the model’s slump distribution.

From the figure, we can see that the slump of all the data is mostly concentrated between 14–24 cm. There are a few outliers below its lower limit, and the overall data are relatively healthy.
(3)Split the dataset as follows: the training set accounts for 70%, the test set accounts for 30% and the feature variables are kept separate from the test set and training set data.(4)Standardized data: There are three main methods for data standardization, namely minimum–maximum standardization, z-score standardization and decimal standardization. This paper adopts the z-score method, also called standard deviation normalization, the mean value of the data processed by it is 0 and the variance is 1. It is currently the most commonly used standardization method. The formula is as follows:
(4)$$x^{*}=\frac{x-\bar{x}}{\sigma }$$
where $\bar{x}$ is the mean value of the corresponding feature and *σ* is the standard deviation.

Part of the processed training data is shown in Table 2:

### 2.4. Model Parameter Selection

The optimization parameters of the slump prediction model considered in this study include the learning rate, momentum factor, the number of hidden nodes in the first layer, the number of hidden nodes in the second layer and the number of iterations. The coefficient of determination (R^2^) and the root mean square error (RMSE) are used as evaluation indicators for the network model. The range values are shown in Table 3, where the number of two-layer and three-layer BP neural network models are 9 × 10 × 10 × 10 and 9 × 10 × 10 × 10 × 10 groups, respectively. Due to the large number and types of parameters, it is quite difficult to find the optimal solution of the model parameters to establish a high-quality concrete slump prediction model. There are three main methods for parameter optimization: algorithm-based, machine-assisted method and manual parameter adjustment [42,43]. Based on the BP neural network algorithm, this study using machine learning as the main method, and optimizing the model parameters by manually adjusting the parameters.

The parameters of each optimization variable are shown in Table 4, where the value of learning rate is 0.1~1.0 and the increment is 0.1; the value of momentum factor is 0.2~1.0, and the increment is 0.1; the value of number of hidden nodes in the first layer is 0.1; the value is 3~12 and the increment is 1; the parameter selection of number of hidden nodes in the second layer is the same as that of number of hidden nodes in the first layer; the value of number of training iterations is 500~5000 and the increment is 500.

## 3. Research on Two-Layer Neural Network

### 3.1. Single Factor Design Experiment of Two-Layer Neural Network

As seen in Table 3, the two-layer neural network involves experimental factors, including learning rate, momentum factor, the number of hidden nodes in the first layer and the number of iterations. Considering that the level of each parameter is large, it will lead to too many trials, and it is not convenient to observe the optimal solution parameter solution. Therefore, this study uses a single-factor experimental design based on the bisection method to obtain the optimal parameter solution without considering the interaction of various factors, aiming to narrow the range of the level of each parameter.

Considering this, parameter optimization is carried out in the established slump model. First, the learning rate, momentum factor and the number of hidden nodes in the first layer are tentatively set to 0.5, 0.6 and 7, respectively, and the optimal number of iterations is obtained, as shown in Table 5. The two parameters of the number of nodes are fixed to obtain the optimal value of the learning rate, and the same processing method is adopted in turn to finally obtain the optimal value of each parameter, as shown in Figure 6.

It can be seen from Figure 6a that RMSE showed a trend of first decreasing and then increasing with the increasing number of iterations, R^2^ showed a trend of first increasing and then decreasing with the increasing number of iterations, and the network performance reached the best when the number of iterations was 3500. Figure 6b–d show that when the learning rate, momentum factor and the number of hidden nodes in the first layer are, respectively 0.2, 0.3 and 4, the network performance is in its best state.

According to this single-factor experimental design, a set of the optimal solutions of parameters without considering interaction are obtained, as shown in Table 6. Thereby narrowing the range of the level value of each factor.

### 3.2. Response Surface Method Test of Two-Layer Neural Network

Response surface optimization method, namely response surface methodology (RSM), is a method for optimizing experimental conditions, which is suitable for solving problems related to nonlinear data processing. Through the regression fitting of the process (as shown in the Equation (5)) and the drawing of the response surface and the contour line, the response value corresponding to each factor level can easily be obtained. On the basis of the response value of each factor level, the predicted optimal response value and the corresponding experimental conditions can be found.
(5)$$Y=B_{0}+\sum_{i}^{k}B_{i}X_{i}+\sum_{i}^{k}B_{ii}X^{2}+\sum_{ij}^{k}B_{ij}X_{i}X_{j}$$
where *Y* represents the response function (in our example, R^2^ and RMSE); *B*_0_ is a constant coefficient; *B_i_*, *B_ii_* and *B_ij_* are the coefficients of linear term, quadratic term [44] and interaction term, respectively.

#### 3.2.1. Model Establishment and Significance Test of a Two-Layer Neural Network

Single factor experiments show that the learning rate, the number of hidden nodes and the number of iterations have a significant impact on network performance. On this basis, R^2^ (*Y*_1_) and RMSE (*Y*_2_) should be taken as the response values and four factors that have a significant impact on network performance: learning rate (*X*_1_), momentum factor (*X*_2_), number of hidden nodes (*X*_3_) and iteration. The number of times (*X*_4_) is the investigation factor. The Box–Behnken test factors and levels are shown in Table 7, and the test results and analysis are shown in Table 8.

When using Design-Expert V10.07 software to fit *Y*_1_ and *Y*_2_ in Table 8, the regression Equation can be obtained:$$\begin{array}{l} Y_{1}=+0.88-0.028X_{1}+1.540\times 10^{-3}X_{2}-9.586\times 10^{-3}X_{3}-4.681\times 10^{-3}X_{4}\\ +9.378\times 10^{-3}X_{1}X_{2}+4.270\times 10^{-3}X_{1}X_{3}-1.483\times 10^{-3}X_{1}X_{4}+9.6\times 10^{-4}X_{2}X_{3}\\ -1.543\times 10^{-3}X_{2}X_{4}+4.247\times 10^{-3}X_{3}X_{4}-1.926\times 10^{-3}X_{1}{}^{2}-6.141\times 10^{-3}X_{2}{}^{2}\\ -2.484\times 10^{-4}X_{3}{}^{2}+5.748\times 10^{-3}X_{4}{}^{2} \end{array}\;\begin{array}{l} Y_{2}=4.20+0.64X_{1}-0.028X_{2}+0.2X_{3}+0.095X_{4}-0.23X_{1}X_{2}-0.094X_{1}X_{3}+0.029X_{1}X_{4}\\ -0.046X_{2}X_{3}+0.02X_{2}X_{4}-0.09X_{3}X_{4}+0.056X_{1}{}^{2}+0.17X_{2}^{2}+0.12X_{3}^{2}-0.092X_{4}^{2} \end{array}$$

The analysis of variance is performed on the above regression equation, and the results are shown in Table 9 and Table 10.

From Table 9 and Table 10, it can be seen that F_1_ = 8.41, F_2_ = 17.83 and the *p* value of *Y*_1_ and *Y*_2_ is less than 0.0001, which indicates that the model is very significant, and the lack of fit term is greater than 0.05, which is not significant. This indicates that the regression equation fits the experiment. The situation is good, and unknown factors have little interference on the experimental results, indicating that the residuals are all caused by random errors. This model has high reliability. After analysis of variance, the primary and secondary order of the influence of the four factors on R^2^ and RMSE is *X*_1_ > *X*_3_ > *X*_4_ > *X*_2_, that is, learning rate > number of hidden nodes > number of iterations > momentum factor. In *Y*_1_, the first-order terms *X*_1_ and *X*_3_ have extremely significant effects on the results; in *Y*_2_, the first-order terms *X*_1_ and *X*_3_ have extremely significant effects on the results, the interaction terms *X*_1 × 2_ have significant effects on the results and the quadratic term *X*_22_ has significant effects on the results.

#### 3.2.2. Response Surface Method Analysis of Two-Layer Neural Network

The response surface curve and contour lines of the interaction of learning rate, momentum factor, number of hidden nodes and number of iterations on R^2^ and RMSE are shown in Figure 7.

The response surface diagram can intuitively indicate the degree of influence of factors on the response value. The more significant the influencing factor, the steeper the slope of the surface. The shape of the contour line can determine the strength of the interaction between the two variables. An ellipse indicates that the interaction between the two factors is significant, and a circle indicates that the interaction between the two factors is not significant.

Figure 7 shows the interaction of learning rate, momentum factor, number of iterations and number of hidden nodes on R^2^ and RMSE. As can be seen from the results, the interaction between the learning rate and the other three factors is the most significant, which is basically consistent with the significance conclusion obtained by the above analysis of variance. The optimal parameters of the model for predicting concrete slump based on a BP neural network are: lr = 0.1, mf = 0.3, noi = 2669 and nohn = 3, where lr stands for learning rate, mf stands for momentum factor, noi stands for number of iterations and nohn stands for the number of hidden nodes. The constructed model with the optimized parameters is used for verification on the training set, and the results are R^2^ = 0.927 and RMSE = 3.373. At the same time this optimized model is used for verification on the test set. After verification, the results are R^2^ = 0.91623 and RMSE = 3.60363. However, the R^2^ of the two-layer baseline model without RSM optimization is only 0.53484509.

#### 3.2.3. Analysis of Slump Influencing Factors Based on Two-Layer Neural Network

Using the optimized parameters to build a two-layer neural network, the training set and the test set were evaluated and verified and the influencing factors of eight variables on the slump were obtained. A bar graph of the effect of each variable on slump is shown in Figure 8.

It can be seen from Figure 8 that the order of influence on slump is flow > coarse aggregate > water > fine aggregate > fly ash > cement > blast furnace slag > superplasticizer. Flow and coarse aggregate are the top two influencing factors with 0.901 and 0.045, respectively. Superplasticizer and blast furnace slag are the two low influencing factors with 0.0006 and 0.0042, respectively.

## 4. Research on Three-Layer Neural Network

### 4.1. Single Factor Design Experiment of Three-Layer Neural Network

The single-factor optimization method of the parameters of the three-layer neural network model is similar to the two-layer neural network and, finally, the optimal value of each parameter is obtained, as shown in Figure 9.

It can be seen from Figure 9a that RMSE and R^2^ show decreasing and increasing trends as the number of iterations increase, and the network performance reaches its best when the number of iterations is 500. Figure 9b–e shows that when the learning rate, momentum factor, the number of hidden nodes in the first layer and the number of hidden nodes in the second layer are, respectively, 0.1, 0.2, 10 and 5, the network performance is in its best state.

According to this single-factor experimental design, a set of optimal solutions of parameters without considering interaction are obtained, as shown in Table 11. Thereby narrowing the range of the level of each factor.

### 4.2. Response Surface Method Test of Three-Layer Neural Network

#### 4.2.1. Model Establishment and Significance Test of Three-Layer Neural Network

Single-factor experiments show that the learning rate, momentum factor, and number of iterations have a significant impact on network performance. On this basis, R^2^ (*Y*_1_) and RMSE (*Y*_2_) can be taken as the response values, and five factors that have a significant impact on network performance (learning rate (*X*_1_), momentum factor (*X*_2_), and number of hidden nodes in the first layer (*X*_3_), the number of hidden nodes in the second layer (*X*_4_) and the number of iterations (*X*_5_)) are the inspection factors. The Box–Behnken test factors and levels are shown in Table 12, and the test results and analysis are shown in Table 13.

Using Design-Expert V10.07 software to fit *Y*_1_ and *Y*_2_ in Table 13, the regression equation can be obtained:$$\begin{array}{l} Y_{1}=+0.88-0.018X_{1}-8.988\times 10^{-3}X_{2}-3.406\times 10^{-3}X_{3}-5.57\times 10^{-3}X_{4}-0.017X_{5}\\ -7.015\times 10^{-3}X_{1}X_{2}+1.9\times 10^{-3}X_{1}X_{3}-0.01X_{1}X_{4}+3.412\times 10^{-3}X_{1}X_{5}-2.757\times 10^{-3}X_{2}X_{3}\\ -4.39\times 10^{-3}X_{2}X_{4}-6.94\times 10^{-3}X_{2}X_{5}+6.12\times 10^{-3}X_{3}X_{4}-4.565\times 10^{-3}X_{3}X_{5}\\ -7.67\times 10^{-3}X_{4}X_{5}+0.013X_{1}{}^{2}+4.512\times 10^{-3}X_{2}{}^{2}+0.012X_{3}{}^{2}+0.011X_{4}^{2}+9.773\times 10^{-3}X_{5}^{2} \end{array}\;\begin{array}{l} Y_{2}=+4.31+0.4X_{1}+0.19X_{2}+0.064X_{3}+0.12X_{4}+0.39X_{5}+0.16X_{1}X_{2}-0.056X_{1}X_{3}\\ +0.23X_{1}X_{4}-0.098X_{1}X_{5}+0.037X_{2}X_{3}+0.88X_{2}X_{4}+0.11X_{2}X_{5}-0.14X_{3}X_{4}+0.11X_{3}X_{5}\\ +0.16X_{4}X_{5}-0.21X_{1}{}^{2}-0.033X_{2}{}^{2}-0.22X_{3}^{2}-0.16X_{4}^{2}-0.14X_{5}^{2} \end{array}$$

The analysis of variance is performed on the above regression equation, and the results are shown in Table 14 and Table 15.

From Table 14 and Table 15, it can be seen that *F*_1_ = 10.21, *F*_2_ = 9.92 and the *p* values of *Y*_1_ and *Y*_2_ are less than 0.0001, which is very significant, and the lack of fit term is greater than 0.05, which is insignificant and shows that the regression equation fits the experiment well. With unknown factors, the interference in the experimental results is small, indicating that the residuals are all caused by random errors and this model has high reliability. After analysis of variance, the primary and secondary order of the five factors on R^2^ and RMSE is *X*_1_ > *X*_5_ > *X*_2_ > *X*_4_ > *X*_3_, that is, learning rate > number of iterations > momentum factor > number of hidden nodes in the second layer > number of hidden nodes in the first layer. In *Y*_1_, the primary terms *X*_1_, *X*_2_, *X*_3_ and *X*_5_ have a significant impact on the results; *X*_4_ has a significant impact on the results; the interaction terms *X*_1_*X*_3_ have a significant impact on the results; the quadratic terms *X*_12_, *X*_32_, *X*_42_ and *X*_52_ have a significant impact on the results. In *Y*_2_, the primary terms *X*_1_, *X*_2_ and *X*_5_ have a significant impact on the results; *X*_4_ has a significant impact on the results; the interaction terms *X*_1_*X*_4_ have a significant impact on the results; the quadratic terms *X*_12_ and *X*_32_ have a significant impact on the results and *X*_42_ and *X*_52_ have a significant impact on the results.

#### 4.2.2. Response Surface Method Analysis of Three-Layer Neural Network

The response surface curve and contour lines of the interaction of learning rate, momentum factor, number of hidden nodes in the first layer, number of hidden nodes in the second layer and number of iterations on R^2^ and RMSE are shown in Figure 10.

As shown in Figure 10, it can be seen from the steepness of the response surface that the learning rate, the number of iterations and the momentum factor have a significant impact on the response value. In contrast, the number of hidden nodes has a much weaker effect on the response value. Most of the contour lines are elliptical, indicating that the interaction between various factors is relatively large.

It can also be seen from Figure 10 that the three-layer neural network used for the verification of the test set has a similar conclusion to the two-layer neural network (optimization model R^2^ > baseline model R^2^, where the optimization model R^2^ = 0.94246, baseline model R^2^ = 0.94246) and the prediction performance of the three-layer neural network for the test set is better than that of the two-layer neural network.

#### 4.2.3. Analysis of Slump Influencing Factors Based on Three-Layer Neural Network

Using a method similar to the two-layer neural network, the influence factors of each variable on the slump in the three-layer neural network model can be obtained. A bar graph of each variable’s effect on slump is also shown in Figure 11.

It can be seen from Figure 11 that the order of influence on slump is flow > water > coarse aggregate > fine aggregate > blast furnace slag > cement > superplasticizer > fly ash. Flow and water are the two top influencing factors with 0.875 and 0.030, respectively. Fly ash and superplasticizer are the two lowest influencing factors with 0.0099 and 0.0101, respectively.

## 5. Conclusions and Analysis

Through the response surface analysis method, the influence of each parameter on the neural network model at different levels is analyzed. As can be seen from the above:(1)In the two-layer neural network model, the learning rate has the most significant impact on the entire model and the change of other parameters has a weaker effect on the network model. The reason for this phenomenon is that as the learning rate increases, the network weight value is updated too much, the swing amplitude exceeds the training range of the model performance and, finally, the system prediction deviation becomes too large. It can also be seen that the network performance of the two-layer neural network is relatively stable. A two-layer neural network constructed by optimization parameters was used to evaluate the test set, and the results are R^2^ = 0.927 and RMSE = 3.373. At the same time, the unoptimized two-layer neural network was evaluated on the test set, and the result was only R^2^ = 0.91623.(2)In the three-layer neural network model, the interaction between the parameters is relatively strong and compared with the two-layer neural network, its predictive ability is stronger. A three-layer neural network constructed by optimization parameters was used to evaluate the test set, and the results are R^2^ = 0.955 and RMSE = 2.781. At the same time, the unoptimized three-layer neural network was evaluated on the test set, and the result was only R^2^ = 0.94246. From the response surface graph, the coefficient of determination and the root mean square error, it can be seen that the three-layer neural network is more stable and more accurate.(3)Interestingly, it can be seen from Figure 8 and Figure 11 that the four main factors affecting the slump are flow, water, coarse aggregate and fine aggregate, which also shows that the two-layer neural network and the three-layer neural network have the same law in evaluating the factors affecting the slump. Of course, there are differences between the two-layer neural network and three-layer neural network in the prediction of influencing factors of the slump. Two-layer neural network results show that coarse aggregate is the second factor affecting slump, while three-layer neural network results indicate that water is the second factor affecting slump. In addition, the influence factor of flow evaluated by the two-layer neural network is even more than 0.9, while the influence factor of each variable evaluated by the three-layer neural network on the slump is relatively reasonable. Therefore, the prediction performance of three-layer neural network is better than that of two-layer neural network.

This paper expounds that the RSM method is used to optimize the parameters of the BP neural network model of concrete slump, and the optimized parameters are used to build the model for the training set and the test set for verification, which are verified to have better performance than the unoptimized benchmark model. However, the research work in this paper still has the following shortcomings:(1)The basic data of concrete slump in this experiment are too small, which, more or less, affects the accuracy of the conclusion. Based on this, it could be considered to further expand the data to achieve a more accurate and reliable effect.(2)The BP neural network used in this paper has a similar “black box” effect, and many model parameters are not interpretable. The next step also requires the use of state-of-the-art deep learning algorithms (e.g., interpretable neural networks) for concrete slump prediction.

## Figures and Tables

**Figure 1 materials-15-04721-f001:**
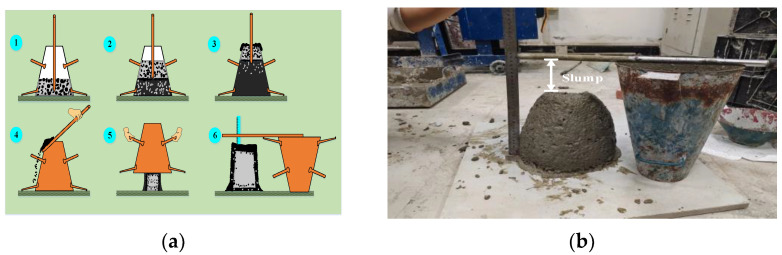
Slump test process, (**a**) schematic diagram of the slump test operation process, step1: first concrete addition, step2: second concrete addition, step3: third concrete addition, step4: smoothing the mouth of the barrel, step5: demoulding, step6: slump measurement; (**b**) schematic diagram of slump test measurement.

**Figure 2 materials-15-04721-f002:**
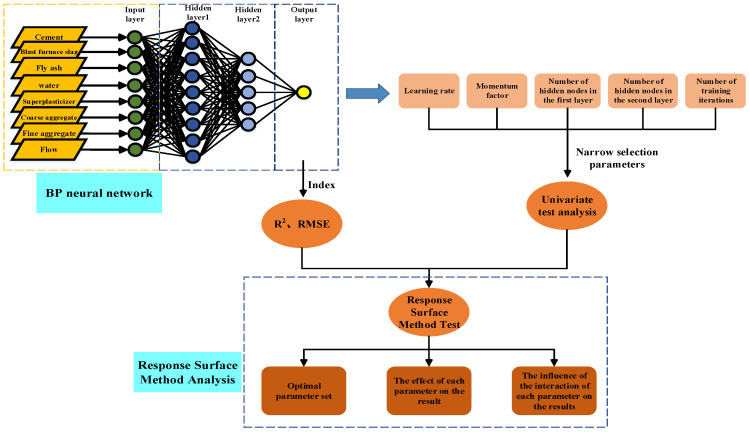
Research architecture diagram.

**Figure 3 materials-15-04721-f003:**
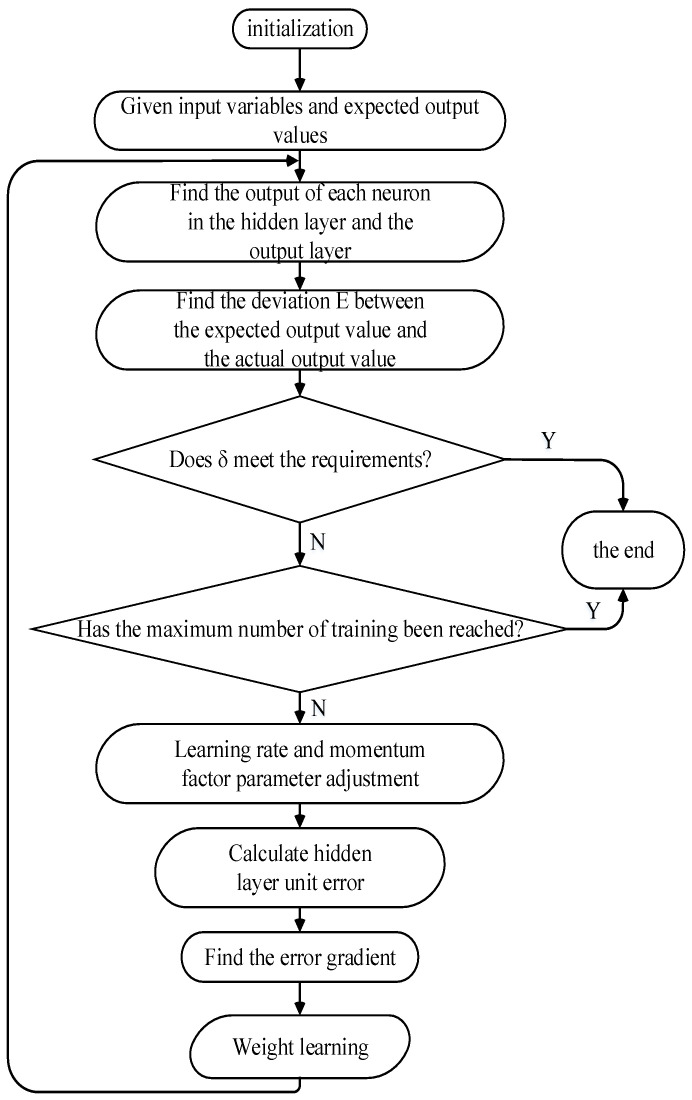
BP neural network algorithm flow chart.

**Figure 4 materials-15-04721-f004:**
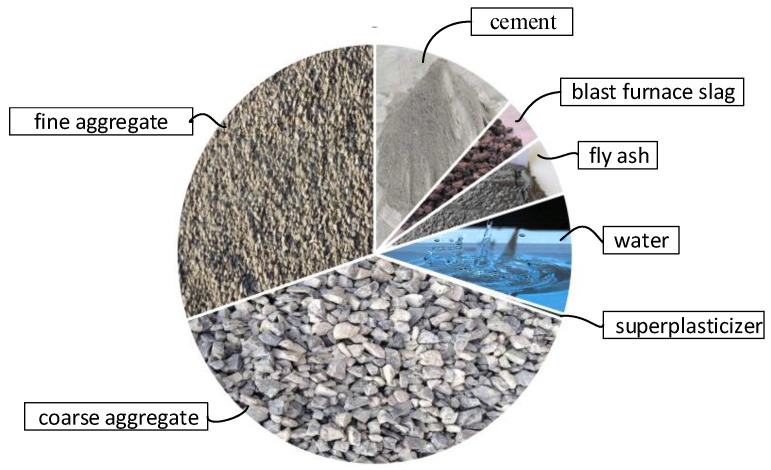
Concrete composition.

**Figure 5 materials-15-04721-f005:**
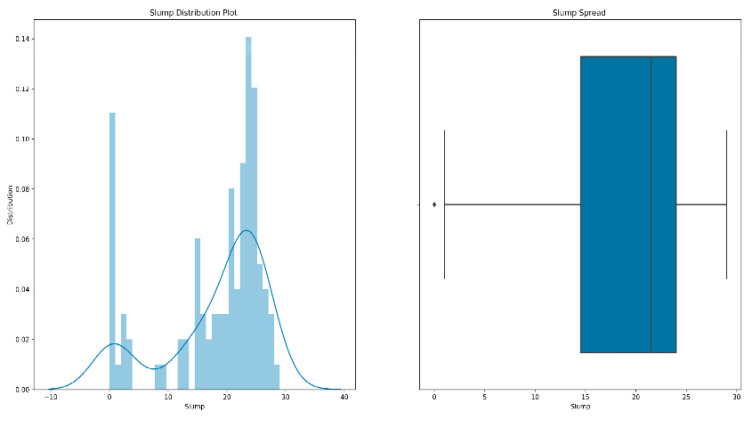
Slump distribution map.

**Figure 6 materials-15-04721-f006:**
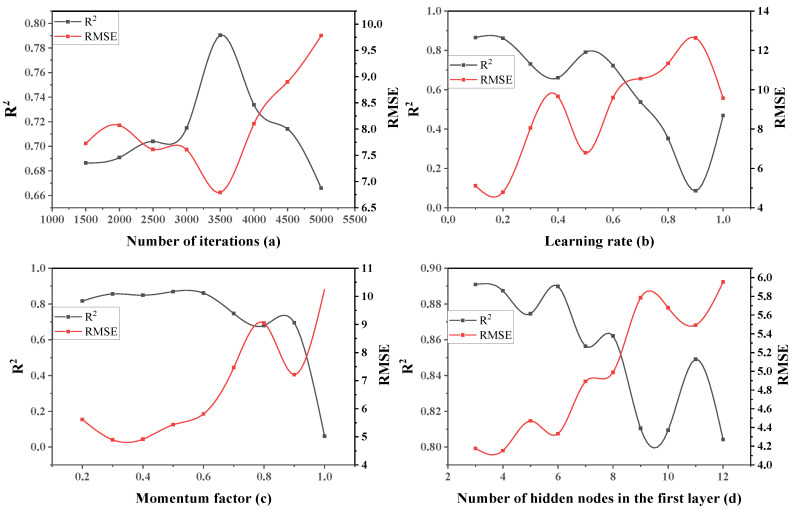
The influence of each single factor on two-layer neural network performance.

**Figure 7 materials-15-04721-f007:**
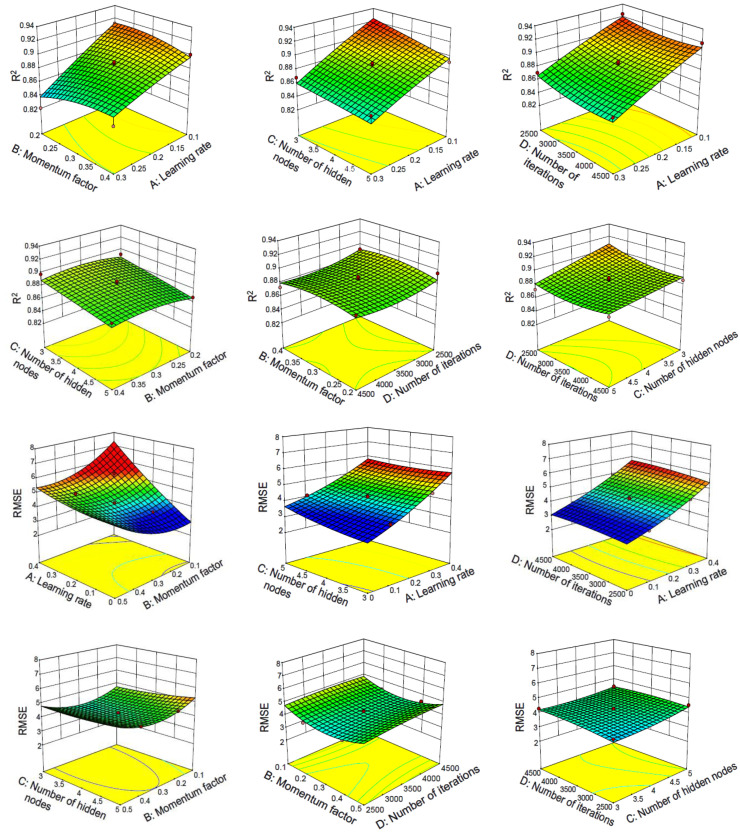
Response surface and contour lines of the interaction of various factors on network performance with two-layer neural network.

**Figure 8 materials-15-04721-f008:**
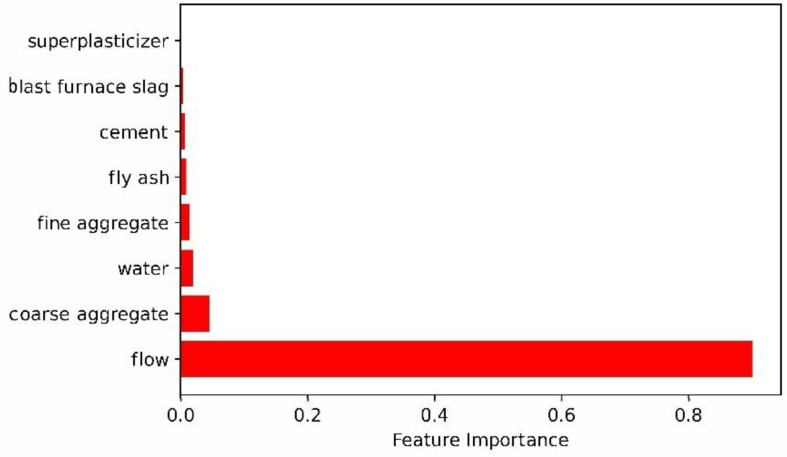
A bar graph of each variable’s effect on slump with two-layer neural network.

**Figure 9 materials-15-04721-f009:**
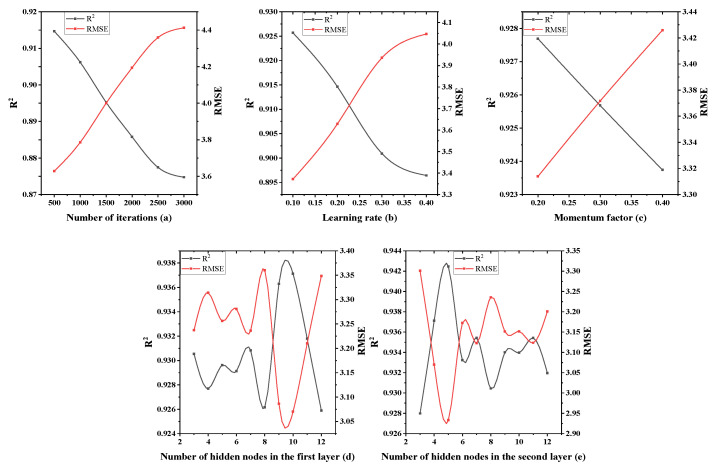
The influence of each single factor on three-layer neural network performance.

**Figure 10 materials-15-04721-f010:**
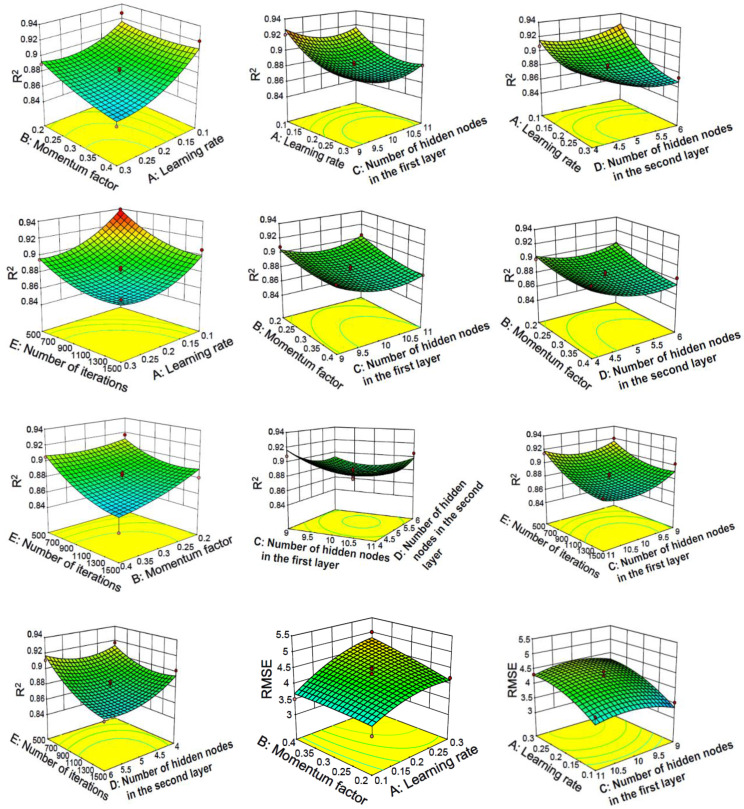
Response surface and contour lines of the interaction of various factors on network performance with three-layer neural network.

**Figure 11 materials-15-04721-f011:**
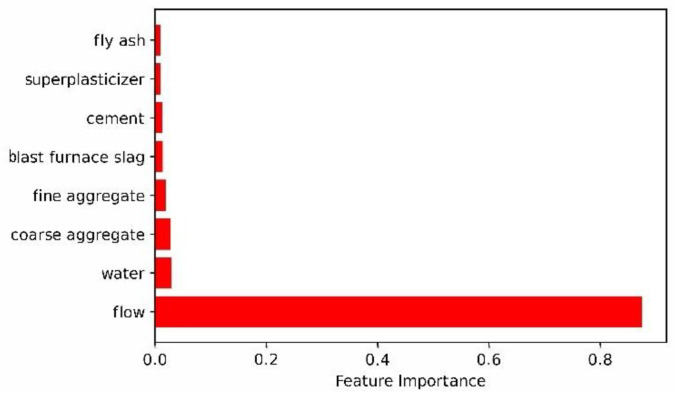
A bar graph of each variable’s effect on slump with three-layer neural network.

**Table 1 materials-15-04721-t001:** Concrete mix ratio.

Mix Ratio	Cement (kg/m^3^)	Blast Furnace Slag (kg/m^3^)	Fly Ash (kg/m^3^)	Water (kg/m^3^)	Superplasticizer (kg/m^3^)	Coarse Aggregate (kg/m^3^)	Fine Aggregate (kg/m^3^)	Flow (mm)	Slump (cm)
C-1	273	82	105	210	9	904	680	62	23
C-2	162	148	190	179	19	838	741	20	1
C-3	147	89	115	202	9	860	829	55	23
C-4	145	0	227	240	6	750	853	58.5	14.5
C-5	148	109	139	193	7	768	902	58	23.75
C-6	374	0	0	190	7	1013	730	42.5	14.5
⋮	⋮	⋮	⋮	⋮	⋮	⋮	⋮	⋮	⋮
C-102	150.3	111.4	238.8	167.3	6.5	999.5	670.5	36.5	14.5
C-103	303.8	0.2	239.8	236.4	8.3	780.1	715.3	78	25

**Table 2 materials-15-04721-t002:** Normalized training set.

Serial Number	Cement	Slag	Fly Ash	Water	SP	Coarse Aggr	Fine Aggr	Flow
98	0.148284	0.417310	1.075807	−1.460561	−0.168038	0.820139	−1.569612	−1.711562
81	−1.206461	−1.099031	0.921458	0.504346	−1.210391	1.567436	−0.873459	0.885664
75	−1.09526	0.553305	−0.112678	−0.243228	−0.800895	0.123258	0.676771	0.561011
53	0.908293	−1.299623	−0.077259	1.019566	0.688181	0.347335	−0.934695	0.767608
46	0.545192	0.264316	−0.362010	0.463936	0.315912	0.022423	−0.950810	0.885664

Note: Slag—blast furnace slag, SP—superplasticizer, Coarse Aggr—coarse aggregate, Fine Aggr—fine aggregate.

**Table 3 materials-15-04721-t003:** Value range of neural network parameters.

Parameter Type	Parameter Selection Range									
learning rate	0.1	0.2	0.3	0.4	0.5	0.6	0.7	0.8	0.9	1.0
momentum factor	0.2	0.3	0.4	0.5	0.6	0.7	0.8	0.9	1.0	-
number of hidden nodes in the first layer	3	4	5	6	7	8	9	10	11	12
number of hidden nodes in the second layer	3	4	5	6	7	8	9	10	11	12
number of training iterations	500	1000	1500	2000	2500	3000	3500	4000	4500	5000

**Table 4 materials-15-04721-t004:** Some examples of batch slump modeling.

Momentum Factor	Learning Rate	Number of Hidden Nodes	Number of Iterations	R^2^	RMSE
0.2	0.1	3	2500	0.862279	3.2887
0.2	0.1	4	2500	0.842614	3.4821
0.2	0.1	5	2500	0.807051	4.0261
0.2	0.1	6	2500	0.763754	4.5256
0.2	0.1	7	2500	0.77447	4.4553
0.2	0.1	8	2500	0.804214	4.1351
0.2	0.1	9	2500	0.771077	4.5278
0.2	0.1	10	2500	0.759843	4.2585
0.2	0.1	11	2500	0.73613	4.66
0.2	0.1	12	2500	0.847652	4.8972

**Table 5 materials-15-04721-t005:** The impact of different iteration times on network performance.

Serial Number	LR	MF	HU	Epochs	R^2^	RMSE
1	0.5	0.6	7	3500	0.79033	6.791
2	0.5	0.6	7	500	0.74777	7.00612
3	0.5	0.6	7	3000	0.71497	7.60734
4	0.5	0.6	7	2500	0.7039	7.61213
5	0.5	0.6	7	1500	0.6864	7.72466
6	0.5	0.6	7	1000	0.65952	8.06336
7	0.5	0.6	7	2000	0.69084	8.07128
8	0.5	0.6	7	4000	0.73372	8.10192
9	0.5	0.6	7	4500	0.71407	8.89968
10	0.5	0.6	7	5000	0.66591	9.78058

**Table 6 materials-15-04721-t006:** Optimal parameter solution under single factor design with two-layer neural network.

Hidden Layers	Number of Hidden Nodes	Learning Rate	Momentum Factor	Number of Training Iterations
1	4	0.2	0.3	3500

**Table 7 materials-15-04721-t007:** Parameter optimization of Box–Behnken test factors and levels with two-layer neural network.

Factor	−1	0	1
Learning rate	0.1	0.2	0.3
Momentum factor	0.2	0.3	0.4
Number of hidden nodes	3	4	5
Number of iterations	2500	3500	4500

**Table 8 materials-15-04721-t008:** Parameter optimization of Box–Behnken test results and analysis with two-layer neural network.

Serial Number	*X* _1_	*X* _2_	*X* _3_	*X* _4_	*Y* _1_	*Y* _2_
1	0.1	0.2	4	3500	0.91166	3.63823
2	0.3	0.2	4	3500	0.82037	5.58347
3	0.1	0.4	4	3500	0.90076	3.90087
4	0.3	0.4	4	3500	0.84698	4.92102
5	0.2	0.3	3	2500	0.89911	4.01026
6	0.2	0.3	5	2500	0.87238	4.529
7	0.2	0.3	3	4500	0.88607	4.27762
8	0.2	0.3	5	4500	0.87633	4.43749
9	0.1	0.3	4	2500	0.92411	3.40633
⋮	⋮	⋮	⋮	⋮	⋮	⋮
28	0.2	0.3	4	3500	0.88984	4.19149
29	0.2	0.3	4	3500	0.87953	4.07428

Note: Design-Expert is a statistical software package from Stat-Ease Inc.

**Table 9 materials-15-04721-t009:** Regression equation coefficients and significance test (R^2^) (two-layer neural network).

Source	Sum of Squares	df	Mean Square	F Value	*p*-Value Prob > F	Significance
Model	0.012	14	8.607 × 10^−4^	8.41	0.0001	significant
*X* _1_	9.547 × 10^−3^	1	9.547 × 10^−3^	93.28	<0.0001	**
*X* _2_	2.846 × 10^−5^	1	2.846 × 10^−5^	0.28	0.6062	
*X* _3_	1.103 × 10^−3^	1	1.103 × 10^−3^	10.77	0.0055	**
*X* _4_	2.629 × 10^−4^	1	2.629 × 10^−4^	2.57	0.1313	
*X* _1 × 2_	3.518 × 10^−4^	1	3.518 × 10^−4^	3.44	0.085	
*X* _1 × 3_	7.293 × 10^−5^	1	7.293 × 10^−5^	0.71	0.4128	
*X* _1 × 4_	8.791 × 10^−6^	1	8.791 × 10^−6^	0.086	0.7738	
*X* _2 × 3_	3.686 × 10^−6^	1	3.686 × 10^−6^	0.036	0.8522	
X_2 × 4_	9.517 × 10^−6^	1	9.517 × 10^−6^	0.093	0.7649	
*X* _3 × 4_	7.217 × 10^−5^	1	7.217 × 10^−5^	0.71	0.4152	
*X* _12_	2.406 × 10^−5^	1	2.406 × 10^−5^	0.24	0.6353	
*X* _22_	2.446 × 10^−4^	1	2.446 × 10^−4^	2.39	0.1444	
*X* _32_	5.259 × 10^−7^	1	5.259 × 10^−7^	5.139 × 10^−3^	0.9439	
*X* _42_	2.143 × 10^−4^	1	2.143 × 10^−4^	2.09	0.1699	
Residual	1.433 × 10^−3^	14	1.024 × 10^−4^			
Lack of Fit	1.335 × 10^−3^	10	1.335 × 10^−4^	5.44	0.0584	not significant
Pure Error	9.810 × 10^−5^	4	2.452 × 10^−5^			

Note: “**” means very significant influence on the result (*p* < 0.01).

**Table 10 materials-15-04721-t010:** Regression equation coefficients and significance test (RMSE) (two-layer neural network).

Source	Sum of Squares	df	Mean Square	F Value	*p*-Value Prob > F	Significance
Model	6.12	14	0.44	17.83	<0.0001	significant
*X* _1_	4.86	1	4.86	198.19	<0.0001	**
*X* _2_	9.479 × 10^−3^	1	9.48 × 10^−3^	0.39	0.5441	
*X* _3_	0.49	1	0.49	20.01	0.0005	**
*X* _4_	0.11	1	0.11	4.39	0.0547	
*X* _1 × 2_	0.21	1	0.21	8.72	0.0105	*
*X* _1 × 3_	0.035	1	0.035	1.44	0.2501	
*X* _1 × 4_	3.39 × 10^−3^	1	3.39 × 10^−3^	0.14	0.7157	
*X* _2 × 3_	8.43 × 10^−3^	1	8.43 × 10^−3^	0.34	0.5669	
*X* _2 × 4_	1.64 × 10^−3^	1	1.64 × 10^−3^	0.067	0.7999	
*X* _3 × 4_	0.032	1	0.032	1.31	0.2711	
*X* _12_	0.02	1	0.02	0.82	0.3798	
*X* _22_	0.18	1	0.18	7.21	0.0178	*
*X* _32_	0.087	1	0.087	3.54	0.081	
*X* _42_	0.055	1	0.055	2.24	0.1566	
Residual	0.34	14	0.025			
Lack of Fit	0.31	10	0.031	4.04	0.0951	not significant
Pure Error	0.031	4	7.72 × 10^−3^			

Note: “*” means significant influence on the result (*p* < 0.05); “**” means very significant influence on the result (*p* < 0.01).

**Table 11 materials-15-04721-t011:** Optimal parameter solution under single factor design with three-layer neural network.

Hidden Layers	Number of Hidden Nodes in the First Layer	Number of Hidden Nodes in the Second Layer	Learning Rate	Momentum Factor	Number of Training Iterations
2	10	5	0.1	0.2	500

**Table 12 materials-15-04721-t012:** Parameter optimization Box–Behnken test factors and levels with three-layer neural network.

Factor	−1	0	1
Learning rate	0.1	0.2	0.3
Momentum factor	0.2	0.3	0.4
Number of hidden nodes in the first layer	9	10	11
Number of hidden nodes in the second layer	4	5	6
Number of iterations	500	1000	1500

**Table 13 materials-15-04721-t013:** Parameter optimization Box–Behnken test results and analysis with three-layer neural network.

Serial Number	*X* _1_	*X* _2_	*X* _3_	*X* _4_	*X* _5_	*Y* _1_	*Y* _2_
1	0.1	0.2	10	5	1000	0.92751	3.34314
2	0.3	0.2	10	5	1000	0.88953	4.18801
3	0.1	0.4	10	5	1000	0.92129	3.50721
4	0.3	0.4	10	5	1000	0.85525	5.00808
5	0.2	0.3	9	4	1000	0.90766	3.8023
6	0.2	0.3	11	6	1000	0.90218	3.93189
7	0.2	0.3	9	4	1000	0.8861	4.26569
8	0.2	0.3	11	6	1000	0.9051	3.82795
9	0.2	0.2	10	5	500	0.91646	3.59023
⋮	⋮	⋮	⋮	⋮	⋮	⋮	⋮
45	0.2	0.3	10	5	1000	0.8715	4.28419
46	0.2	0.3	10	5	1000	0.8857	4.32548

**Table 14 materials-15-04721-t014:** Regression equation coefficients and significance test (R^2^) (three-layer neural network).

Source	Sum of Squares	df	Mean Square	F Value	*p*-Value Prob > F	Significance
Model	0.016	20	7.88 × 10^−4^	10.21	<0.0001	significant
*X* _1_	5.323 × 10^−3^	1	5.323 × 10^−3^	68.99	<0.0001	**
*X* _2_	1.293 × 10^−3^	1	1.293 × 10^−3^	16.75	0.0004	**
*X* _3_	1.856 × 10^−4^	1	1.856 × 10^−4^	2.41	0.1335	**
*X* _4_	4.964 × 10^−4^	1	4.964 × 10^−4^	6.43	0.0178	*
*X* _5_	4.399 × 10^−3^	1	4.399 × 10^−3^	57.02	<0.0001	**
*X* _1_ *X* _2_	1.968 × 10^−4^	1	1.968 × 10^−4^	2.55	0.1228	
*X* _1_ *X* _3_	1.444 × 10^−5^	1	1.444 × 10^−5^	0.19	0.669	
*X* _1_ *X* _4_	4.19 × 10^−4^	1	4.19 × 10^−4^	5.43	0.0281	*
*X* _1_ *X* _5_	4.658 × 10^−5^	1	4.658 × 10^−5^	0.6	0.4444	
*X* _2_ *X* _3_	3.042 × 10^−5^	1	3.042 × 10^−5^	0.39	0.5358	
*X* _2_ *X* _4_	7.709 × 10^−5^	1	7.709 × 10^−5^	1	0.3271	
*X* _2_ *X* _5_	1.927 × 10^−4^	1	1.927 × 10^−4^	2.5	0.1266	
*X* _3_ *X* _4_	1.498 × 10^−4^	1	1.498 × 10^−4^	1.94	0.1757	
*X* _3_ *X* _5_	8.336 × 10^−5^	1	8.336 × 10^−5^	1.08	0.3086	
*X* _4_ *X* _5_	2.353 × 10^−4^	1	2.353 × 10^−4^	3.05	0.093	
*X* _1_ ^2^	1.451 × 10^−3^	1	1.451 × 10^−3^	18.81	0.0002	**
*X* _2_ ^2^	1.776 × 10^−4^	1	1.776 × 10^−4^	2.3	0.1417	
*X* _3_ ^2^	1.326 × 10^−3^	1	1.326 × 10^−3^	17.19	0.0003	**
*X* _4_ ^2^	9.629 × 10^−4^	1	9.629 × 10^−4^	12.48	0.0016	**
*X* _5_ ^2^	8.336 × 10^−4^	1	8.336 × 10^−4^	10.8	0.003	**
Residual	1.929 × 10^−3^	25	7.715 × 10^−5^			
Lack of Fit	1.707 × 10^−3^	20	8.535 × 10^−5^	1.92	0.2418	not significant
Pure Error	2.219 × 10^−4^	5	4.438 × 10^−5^			

Note: “*” means significant influence on the result (*p* < 0.05); “**” means very significant influence on the result (*p* < 0.01).

**Table 15 materials-15-04721-t015:** Regression equation coefficients and significance test (RMSE) (three-layer neural network).

Source	Sum of Squares	df	Mean Square	F Value	*p*-Value Prob > F	Significance
Model	7.35	20	0.37	9.92	<0.0001	significant
*X* _1_	2.6	1	2.6	70.13	<0.0001	**
*X* _2_	0.56	1	0.56	15.02	0.0007	**
*X* _3_	0.066	1	0.066	1.79	0.1933	
*X* _4_	0.22	1	0.22	5.84	0.0233	*
*X* _5_	2.49	1	2.49	67.26	<0.0001	**
*X* _1_ *X* _2_	0.11	1	0.11	2.9	0.1009	
*X* _1_ *X* _3_	0.013	1	0.013	0.34	0.5633	
*X* _1_ *X* _4_	0.21	1	0.21	5.6	0.026	*
*X* _1_ *X* _5_	0.038	1	0.038	1.03	0.3192	
*X* _2_ *X* _3_	5.456 × 10^−3^	1	5.456 × 10^−3^	0.15	0.7046	
*X* _2_ *X* _4_	0.031	1	0.031	0.83	0.372	
*X* _2_ *X* _5_	0.05	1	0.05	1.34	0.258	
*X* _3_ *X* _4_	0.08	1	0.08	2.17	0.1532	
*X* _3_ *X* _5_	0.049	1	0.049	1.31	0.2626	
*X* _4_ *X* _5_	0.1	1	0.1	2.73	0.1109	
*X* _1_ ^2^	0.38	1	0.38	10.32	0.0036	**
*X* _2_ ^2^	9.445 × 10^−3^	1	9.445 × 10^−3^	0.25	0.6182	
*X* _3_ ^2^	0.41	1	0.41	10.99	0.0028	**
*X* _4_ ^2^	0.22	1	0.22	5.88	0.0229	*
*X* _5_ ^2^	0.17	1	0.17	4.47	0.0446	*
Residual	0.93	25	0.037			
Lack of Fit	0.87	20	0.044	3.96	0.0664	not significant
Pure Error	0.055	5	0.011			

Note: “*” means significant influence on the result (*p* < 0.05); “**” means very significant influence on the result (*p* < 0.01).

## Data Availability

Not applicable.

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
