# Peer review of "Research on Hyperparameter Optimization of Concrete Slump Prediction Model Based on Response Surface Method"

_materials, 2022, doi:10.3390/ma15134721_

Round 1

Reviewer 1 Report

Please incorporate the following major changes before the research study is considered for publication:

  1. The introduction section need a lot of improvement and justification used in first two paragraphs are not quite true as the slump test is the most widely, easy and reliable test used globally.
  2. The reference citing is too confusing. It is difficult to even read the actual text of the paper.
  3. The data set used in the research is experimental data and source of this data is not mentioned.
  4. The discussion of the test results should incorporate the effect of different variables like effect of flyash, coarse aggregate etc ( increase or decrase) on the slump and complement it with existing literature rather than discussing the model parameters.
  5. How the developed model will help the construction industry in specific and its practical application should be highlighted.
  6. Please explain how the validation of the model was done against experimental data .  

Author Response

Point 1:The introduction section need a lot of improvement and justification used in first two paragraphs are not quite true as the slump test is the most widely, easy and reliable test used globally.

Response 1: Thank you for your suggestion, according to your suggestion, in the revised manuscript, we have made changes in the introduction, see page 1 line 31 to page 3 line 8 of the revised manuscript.

Point 2:The reference citing is too confusing. It is difficult to even read the actual text of the paper.

Response 2:Thanks for your opinion. In the revised manuscript, the reference citation format has been modified to match your journal's format.

Point 3:The data set used in the research is experimental data and source of this data is not mentioned.

Response 3:Thank you for your suggestion, based on your suggestion, in the revised manuscript, the source of the data has been explained in Section 2.2, page 5, line 5.

Point 4:The discussion of the test results should incorporate the effect of different variables like effect of flyash, coarse aggregate etc ( increase or decrase) on the slump and complement it with existing literature rather than discussing the model parameters.

Response 4:Thank you for your suggestion. According to your suggestion, in the revised manuscript, the relationship between variables is not discussed in this paper. For this reason, the fifth point of Section 2.3, which involves the discussion of variable relationship, has been deleted.

Point 5:How the developed model will help the construction industry in specific and its practical application should be highlighted.

Response 5:Thank you for your suggestion, and based on your suggestion, in the revised manuscript, I have added corresponding text to the Abstract, Introduction, and Conclusion sections to address this issue.

Point 6:Please explain how the validation of the model was done against experimental data.

Response 6:Thanks for your opinion. In the revised manuscript, introduced in Section 2.3 and the conclusion, 103 sets of data are split into training and test sets, of which the training set accounts for 70% and the test set accounts for 30%. In this experiment, parameter optimization research is carried out on the training set, and the final optimal model is verified on the test set.

Reviewer 2 Report

The authors developed a back propagation neural network to predict the concrete slump using 103 experimental test data from the literature. The parameters were optimized using the response surface methodology. However, there could be a more rigorous interpretation and scientific explanations of the mechanisms behind the observed trends. So, it is recommended to elaborate more on the results. In addition, I believe that the below mentioned comments should also be addressed to strengthen the manuscript.

General comment:

1) There are a lot of grammatical and punctuation errors that must be addressed. For example:

  • “Sigmod function” should be replaced by “Sigmoid function” throughout the manuscript.
  • “R2” and “R^2” should be replaced by “R2

2) Many sentences in the manuscript read like a step-by-step tutorial rather than a technical paper write-up. This must be fixed. For example:

  • In Page 9: Now adjust the selected network parameters in the table in order to output the evaluation index of the prediction result of the neural network model, and realize the large-scale accurate modeling.
  • In Page 10: Similarly, fix the optimal number of iterations with the other two parameters to obtain the optimal value of a set of single factors.

3) The in-text citation format should be changed. The current format is too long and confusing.

In Introduction: Authors stated that Slump is an important indicator to measure the homogeneity of concrete quality, which is reflected in the fact that excessive slump will slow down the hardening speed of concrete and low strength after hardening, which greatly affects the quality of the project.” & Precise control of slump is a prerequisite to ensure excellent performance of concrete.” Please add references. Some suggested references are: "Influence of carbon nanotubes on SCC flowability", and "Carbon nanotube reinforced cementitious composites: A comprehensive review."

The introduction has no proper of focus, and the novelty and knowledge gap of study are not stated. This should be improved. For example, although machine learning algorithms could be used to predict the properties of the concrete (e.g., slump, compressive strength, etc.), there is no physical meaning in machine learning algorithms. So, I suggest that the authors explain the limitations of the machine learning algorithms and alternative options to overcome these limitations. See these references to discuss this: "Design and Predicting performance of carbon nanotube reinforced cementitious materials: mechanical properties and dispersion characteristics", "Modeling the mechanical properties of cementitious materials containing CNTs", and "Probabilistic model for flexural strength of Cementitious Materials Containing CNTs."

Figure 3 is a part of the Figure 2. There is no point to add a separate Figure 3!

In Section 2.2; Page 6: Authors stated thatThe concrete matching ratio is shown in Table 1 and Figure 5.” What is the “matching ratio?” Do you mean the mix proportion?

In Section 2.2; Page 6: “Among them, cement, blast furnace slag, fly ash, water, superplasticizer, coarse aggregate, fine aggregate and fluidity are the inputs of neural network variables, and the slump is the output of variables.” In this sentence the authors mentioned “fluidity” as one of the inputs. In Table 1, it is called “Liquidity” and in Figure 2 it is called “Mobility.” Please be consistent! Also, what is the fluidity and how it is different from slump. This should be explained in the manuscript.

In Network Data Preprocessing: Please add references when you explain data shuffling and splitting. The following references can be used: "Probabilistic model for flexural strength of carbon nanotube reinforced cement-based materials" and "Elastic modulus formulation of cementitious materials incorporating carbon nanotubes: Probabilistic approach."

In the Outlier processing: What was the criteria for data outlier selection? For example, in this reference "Mechanical properties of carbon-nanotube-reinforced cementitious materials: database and statistical analysis," a data point is considered to be an outlier if it lies 1·5 times the interquartile range below the first quartile or above the third quartile. Please use the reference to explain this.

In Figures 7-11: The font size is too small to read. Please increase the font size.

In Page 9; Lines 10-11:Therefore, the amount of cement and water are the characteristics that have the highest correlation with slump.” What about the aggregate?

In Page 9; Lines 12-13: In contrast, there is no significant linear correlation between slag and compressive strength.” Should it be “compressive strength” or “slump?”

In 2.4 Model parameter selection; Lines 6-8:The range values are shown in Table 3, where the number of two-layer and three-layer BP neural network models are 90,000 and 900,000 groups, respectively.” Please explain how you get these values from Table 3?

In Page 11: “It can be seen from Figure 8(a) that RMSE and R2 show a trend of first decreasing and then increasing, and first increasing and then decreasing with the increasing number of iterations.” Actually, RSME exhibits a decreasing trend at first followed by an increasing trend!

In Page 12; Lines 1-2: Authors stated that Through the regression fitting of the process (as shown in the formula) and the drawing of the response surface and the contour line, …” Please mention as shown in Equation 5!

In Tables 9, 14, and 15: Define “*” and “**” in the tables. These are only explained in Table 10!

In Page 16: Figure 10(b), 10(c), 10(d) and 10(e) shows that when the learning rate, momentum factor, the number of hidden nodes in the first layer, and the number of hidden nodes in the second layer are respectively 0.1, 0.3, 10, and 5, the network performance is in the best state.” Isn’t 0.2 the best value in Figure 10(c)?

Author Response

Point 1: There are a lot of grammatical and punctuation errors that must be addressed. For example:

“Sigmod function” should be replaced by “Sigmoid function” throughout the manuscript.

“R2” and “R^2” should be replaced by “R2

Response 1: Thanks for your opinion. In the revised manuscript, revisions have been made. See page 4 of the revised manuscript (lines 14, 15, 24). Section 2.4 line 6 on page 7, Table 5 on page 9, Line 3 on page 10, Section 3.2 (Lines 14, 22, 24, 25), Table 8 on page 11, Page 12 (Lines 8, 9, 16, 18-21, 27)

Point 2: Many sentences in the manuscript read like a step-by-step tutorial rather than a technical paper write-up. This must be fixed. For example:

In Page 9: Now adjust the selected network parameters in the table in order to output the evaluation index of the prediction result of the neural network model, and realize the large-scale accurate modeling.

In Page 10: Similarly, fix the optimal number of iterations with the other two parameters to obtain the optimal value of a set of single factors.

Response 2: Thanks for your opinion. In the revised manuscript, I have checked the entire text for such issues and made corrections to the corresponding issues.

Point 3: The in-text citation format should be changed. The current format is too long and confusing.

Response 3: Thanks for your opinion. In the revised manuscript, the in-text citation format has been modified.

Point 4: In Introduction: Authors stated that “Slump is an important indicator to measure the homogeneity of concrete quality, which is reflected in the fact that excessive slump will slow down the hardening speed of concrete and low strength after hardening, which greatly affects the quality of the project.” & “Precise control of slump is a prerequisite to ensure excellent performance of concrete.” Please add references. Some suggested references are: "Influence of carbon nanotubes on SCC flowability", and "Carbon nanotube reinforced cementitious composites: A comprehensive review."

Response 4: Thanks for your opinion. In the revised manuscript, a reference has been added to page 2, line 2.

Point 5: The introduction has no proper of focus, and the novelty and knowledge gap of study are not stated. This should be improved. For example, although machine learning algorithms could be used to predict the properties of the concrete (e.g., slump, compressive strength, etc.), there is no physical meaning in machine learning algorithms. So, I suggest that the authors explain the limitations of the machine learning algorithms and alternative options to overcome these limitations. See these references to discuss this: "Design and Predicting performance of carbon nanotube reinforced cementitious materials: mechanical properties and dispersion characteristics", "Modeling the mechanical properties of cementitious materials containing CNTs", and "Probabilistic model for flexural strength of Cementitious Materials Containing CNTs."

Response 5: Thank you for your suggestion, the Introduction section has been revised in the revised manuscript according to your suggestion.

Point 6: Figure 3 is a part of the Figure 2. There is no point to add a separate Figure 3!

Response 6: Thanks for your opinion. In the revised manuscript, Figure 3 has been removed.

Point 7: In Section 2.2; Page 6: Authors stated that “The concrete matching ratio is shown in Table 1 and Figure 5.” What is the “matching ratio?” Do you mean the mix proportion?

Response 7: We thank the reviewer for pointing out this issue. In Section 2.2, page 5, line 12, the misunderstanding due to typo has been revised.

Point 8: In Section 2.2; Page 6: “Among them, cement, blast furnace slag, fly ash, water, superplasticizer, coarse aggregate, fine aggregate and fluidity are the inputs of neural network variables, and the slump is the output of variables.” In this sentence the authors mentioned “fluidity” as one of the inputs. In Table 1, it is called “Liquidity” and in Figure 2 it is called “Mobility.” Please be consistent! Also, what is the fluidity and how it is different from slump. This should be explained in the manuscript.

Response 8: Thanks for your opinion. In the revised manuscript, in section 2.2, page 5, line 14, revisions and additional explanations have been made.

Point 9: In Network Data Preprocessing: Please add references when you explain data shuffling and splitting. The following references can be used: "Probabilistic model for flexural strength of carbon nanotube reinforced cement-based materials" and "Elastic modulus formulation of cementitious materials incorporating carbon nanotubes: Probabilistic approach."

Response 9: Thanks for your opinion. In the revised manuscript, in section 2.3, data shuffling and splitting references have been added.

Point 10: In the Outlier processing: What was the criteria for data outlier selection? For example, in this reference "Mechanical properties of carbon-nanotube-reinforced cementitious materials: database and statistical analysis," a data point is considered to be an outlier if it lies 1·5 times the interquartile range below the first quartile or above the third quartile. Please use the reference to explain this.

Response 10 : Thanks for your opinion. In the revised manuscript, in section 2.3, line 14, the criteria for selection of data outliers have been explained.

Point 11: In Figures 7-11: The font size is too small to read. Please increase the font size.

Response 11: Thanks for your opinion. In the revised manuscript, the font issues in Figures 6-9 have been revised.

Point 12: In Page 9; Lines 10-11: “Therefore, the amount of cement and water are the characteristics that have the highest correlation with slump.” What about the aggregate?

Response 12: We thank the reviewer for pointing out this issue. This paper does not discuss the relationship between variables, for this reason, the fifth point of Section 2.3, which involves the discussion of variable relationship, has been deleted.

Point 13: In Page 9; Lines 12-13: “In contrast, there is no significant linear correlation between slag and compressive strength.” Should it be “compressive strength” or “slump?”

Response 13: We thank the reviewer for pointing out this issue. Section 2.3 fifth point deleted.

Point 14: In 2.4 Model parameter selection; Lines 6-8: “The range values are shown in Table 3, where the number of two-layer and three-layer BP neural network models are 90,000 and 900,000 groups, respectively.” Please explain how you get these values from Table 3?

Response 14: Thanks for your opinion. In the revised manuscript, regarding model parameter selection, the data writing errors of 90,000 and 900,000 have been revised for the first error. Secondly, explain how to obtain these values from Table 3: Arrange and combine within the value range according to each model parameter, a total of 9×10×10×10 = 9000 and 9×10×10×10×10 = 90000 group data.

Point 15: In Page 11: “It can be seen from Figure 8(a) that RMSE and R2 show a trend of first decreasing and then increasing, and first increasing and then decreasing with the increasing number of iterations.” Actually, RSME exhibits a decreasing trend at first followed by an increasing trend!

Response 15: Thanks for your opinion. In the revised manuscript, page 10, the description error for Figure 6(a) has been corrected.

Point 16: In Page 12; Lines 1-2: Authors stated that “Through the regression fitting of the process (as shown in the formula) and the drawing of the response surface and the contour line, …” Please mention as shown in Equation 5!

Response 16: Thanks for your opinion. In the revised manuscript, page 10, section 3.2, line 5 has been revised.

Point 17: In Tables 9, 14, and 15: Define “*” and “**” in the tables. These are only explained in Table 10!

Response 17: Thanks for your opinion. In the revised manuscript, Table 9, Table 14 and Table 15 have been supplemented.

Point 18: In Page 16: “Figure 10(b), 10(c), 10(d) and 10(e) shows that when the learning rate, momentum factor, the number of hidden nodes in the first layer, and the number of hidden nodes in the second layer are respectively 0.1, 0.3, 10, and 5, the network performance is in the best state.” Isn’t 0.2 the best value in Figure 10(c)?

Response 18: We thank the reviewer for pointing out this issue. On page 14, Figure 8 description error has been modified.

Reviewer 3 Report

Your article lacks presentation, you should look at the comments carefully. 

Author Response

Point 1: It is necessary to introduce some interesting results that you have found in the abstract.
Response 1: Thanks for your opinion. In the revised manuscript, the abstract has been revised accordingly.
Point 2: Do not put the references in the text. Redo your article by separating the references from the text. Give only reference numbers in the text. References are at the end of the manuscript.
Response 2: Thanks for your opinion. In the revised manuscript, the references have been revised.
Point 3: Tables 14 and 15 do not have the same typeface and font as the other tables in the article.
Response 3: We thank the reviewer for pointing out this issue. Tables 14 and 15 fonts have been modified.
Point 4: In the conclusion of an article, you do not analyze, you give results and prospects of your research work. 
Conclusion and prospects
Response 4: Thanks for your opinion. In the revised manuscript, the conclusion part has explained and prospected the shortcomings of this paper.

Round 2

Reviewer 1 Report

Thank you for replying to the reviewers comments and trying to modify the manuscript, however, the reviewer is of the opinion that, the paper still lacks the basic link between the modeling results and physical interpretation with regard to civil/construction engineering. The paper is current form is not well suited for the materials journals and is of very less interest for civil/construction audience, therefore, I will recommend the authors to target some automation based journals.

Author Response

Thank you very much for your valuable suggestions. The revised edition adds a study of factors influencing slump.

Reviewer 2 Report

The authors have satisfactorily addressed the reviewer's comments. 

Author Response

Thank you for your suggestion.